# Secure Inter-Domain Routing Based on Blockchain: A Comprehensive Survey

**DOI:** 10.3390/s22041437

**Published:** 2022-02-13

**Authors:** Lukas Mastilak, Pavol Helebrandt, Marek Galinski, Ivan Kotuliak

**Affiliations:** Faculty of Informatics and Information Technologies, Slovak University of Technology in Bratislava, Ilkovicova 2, 842 16 Bratislava, Slovakia; pavol.helebrandt@stuba.sk (P.H.); marek.galinski@stuba.sk (M.G.); ivan.kotuliak@stuba.sk (I.K.)

**Keywords:** inter-domain routing, border gateway protocol, network security, blockchain

## Abstract

The whole Internet consists of thousands of autonomous systems that transfer data with one another. The BGP plays a significant role in routing, but its behaviour is essentially naive, trusting neighbours without authenticating advertised IP prefixes. This is the main reason why BGP endures various path manipulation attacks. Recently, conventional methods for securing BGP have been implemented, i.e., BGPSec with RPKI. However, these approaches are centralised with a single point of failure that may be compromised, invalidating the whole security mechanism. There have been multiple decentralised projects dealing with various mechanisms, mostly built on Ethereum and blockchain networks. Some with ambition to strengthen existing centralised mechanisms, others to replace them. In this article, we present the first comprehensive survey on blockchain solutions to enforce BGP security, with complex explanations of their contributions and a comparison with different aspects. We explain how blockchain technology can provide an alternative to prevent the false origin of IP prefixes or hijacking AS paths. Moreover, we describe new blockchain-based attacks that BGP would face after the inclusion of blockchain into the inter-domain routing. Finally, we answer the defined research questions and discuss the potential open issues for further study.

## 1. Introduction

The whole Internet consists of thousands of networks called autonomous systems. The data are transferred between them through paths that are determined by the routing process. A single administrative entity independently manages the autonomous system. The Border Gateway Protocol (BGP) enables the exchange of routing information among the autonomous systems to create paths for transferring data [1]. This process is called inter-domain routing.

In recent years, the distributed ledger called blockchain has became a popular technology thanks to Bitcoin cryptocurrency. It introduced peer-to-peer payments in the digital world. Since then, hundreds of projects have been built on blockchain, for example, Ethereum that brought smart contract (SC), the self-executing code with terms of the agreement between a sender and a receiver. Generally, before we decide to deploy blockchain technology, we should meet the following assumptions [2]:The data will be shared among untrusted participants;The participants will maintain a blockchain system, not a central authority;The participants will require public and immutable proof for the performed transactions.

Inter-domain routing satisfies those assumptions for using blockchain. There are autonomous systems that do not trust each other, Border Gateway Protocol (BGP) offers almost no authentication of the content of received BGP messages, and the deployment of Resource Public Key Infrastructure (RPKI) is progressing relatively slowly. Autonomous systems often fight against attacks, such as BGP hijacking or route leaks, that affect the incorrect spread of routing information, which can influence the availability of services [3]. Blockchain technology can be an alternative or supplementary system to the existing structures to improve the authentication of the routing information in the inter-domain environment. Despite the attractive characteristics of blockchain, one of the critical limitations of blockchain is scalability, which can be an obstacle in its deployment [4]. The number of transactions processed per second has to be greater than or equal to the existing structure.

In this work, we describe existing improvements in BGP security and well-known BGP attacks. We explain how blockchain technology can provide an alternative in order to prevent the false origin of IP prefixes or hijacking AS paths. Moreover, we describe new blockchain-based attacks that BGP would face after incorporating blockchain into the inter-domain routing. Furthermore, we present an overview of the literature on blockchain that enhances BGP security. As such, we summarise and discuss autonomous system (AS) path protection and limitations in the related literature. On top of this, we analyse the scalability and limitations of blockchain in the context of inter-domain routing.

This article presents the first comprehensive survey of blockchain-based solutions for increasing inter-domain routing security of BGP. It enhances systematisation of knowledge in this novel area and suggests possible directions for future research.

### 1.1. Contributions

Our contributions are summarised as follows:We describe the security of BGP and provide a general overview of existing BGP attacks;We present the benefits of blockchain technology and smart contracts as well as their possible drawbacks;We summarise the recent works on blockchain that enhance the security of BGP, and we discuss their technical characteristics and limitations;We analyse the protection of the AS path against any modification, and we investigate the scalability of those implementations;We answer the defined research questions, and we mention the potential open issues for improvement of the field.

### 1.2. Outline

The rest of the paper is organised as follows. In Section 2, we discuss the concept of BGP and its implemented security mechanisms to protect against potential threats and review existing attacks. In Section 3, we discuss blockchain and the threats which it must resist. In Section 4, we introduce the approach we adhered to while selecting the works to be included in our review and formulate research questions. Section 5 provides short descriptions of all included works and introduces thematic categories. The analysis we prepared for answering the formulated research questions is in Section 6, Section 7, Section 8 and Section 9. Reflections and open research issues are discussed in Section 10. Section 11 concludes the paper.

## 2. Overview of BGP

The Internet is a network that has been comprised of thousands of smaller networks. These are called autonomous systems (ASes) and exchange routing information amongst them. This task is accomplished using BGP. As a result, this protocol is responsible for looking at the best path between nodes with different geographical locations. An AS is a large pool of routers administrated by a single organisation. The AS announces its own IPv4/IPv6 prefixes to neighbours and may provide transit services for other ASes. It means that when an AS receives prefix from a neighbour, it will announce it to other neighbours in compliance with its internal policies. The routing of packets is managed by internal BGP (iBGP) within the AS. The communication between neighbouring ASes is ensured by external BGP (eBGP). In every BGP router is a BGP table which maintains the current best path for each learned prefix. The selection of the path is based on the list of attributes that are attached to the prefixes. One of the critical attributes is the AS path used to prevent loops in BGP. The AS path is a series of ASes that a specific route passes through to reach one router. If the AS detects its own AS number in the AS path, it will reject the prefix. In BGP, the best path still does not mean the shortest path because ASes often prefer their own routing strategy.

The BGP [5] was developed in 1994 when requirements on security were lower than today. In general, BGP has naive behaviour because it trusts its neighbours without authenticating advertised IP prefixes. As a result, an AS can announce illegitimate IP prefixes that it does not own. Thus, these false announcements can cause a limitation of the availability of services on the Internet. Furthermore, some ASes can even replace a path in the BGP table with a false path that modifies the packets’ forwarding.

### 2.1. BGP Attacks

BGP has to face various path manipulation attacks. A lack of an announcement verification causes security risks such as prefix hijacking or a route leak. In this section, we discuss these attacks.

#### 2.1.1. Route Leak

In most cases, a route leak is not an attacker’s planned activity, but it is caused by misconfiguration. An administrator intentionally or unintentionally breaks a routing policy that modifies the propagation of routes based on business relationships between ASes. According to Giotsas [6], there are three categories of business relationships: provider-to-customer, peer-to-peer, and sibling-to-sibling. If the AS violates route policies when it propagates routes, it may announce a non-existent path or create a loop. In RFC 7908 [7], there are six types of route leaks, but some of them can be eliminated by RPKI. As a result, the neighbour AS can prefer a non-existent shorter path to the original path. In the past, several detection mechanisms [8,9] of route leaks were designed to resolve this problem.

#### 2.1.2. BGP Hijacking

One of the most serious network attacks is BGP Hijacking. An attacker tries to steal IP prefixes and reroute traffic onto incorrect exits. For this attack to occur, the attacker has to compromise an AS and announce fake IP prefixes that the attacked AS does not own. In a less likely scenario, the provider intentionally sends false announcements to their neighbours. The primary aim may be to make part of services unreachable or gain control over the transit of the victim. Overall, BGP protocol trusts neighbour ASes and hence does not verify the legitimacy of received messages. As a result, the false learned route will be quickly spread on the Internet. The BGP routers select the best path according to the priority of BGP attributes into its BGP table. However, they can easily accept a false route with better attributes than the original route. This attack has two types: partial attack and complete attack. In the case of a partial attack, the attacker announces a prefix with the same mask as the original prefix but with better attributes. The complete attack propagates a prefix with a more specific mask compared to the original prefix [10,11]. In [12], they classify BGP events as follows:Typos;Prepending mistakes;Origin changes;Forged AS paths.

Data from Cisco’s BGPStream [13], a public service platform providing information about route events, showed that the number of hijack incidents in 2020 increased from 2019. The total number of incidents in 2020, BGP hijacking and BGP leaks, dropped from 4202 incidents in 2019 to 3873 incidents.

The analysis of BGP attacks [14] in the last two years shows that BGP hijacking has increased. The comparison is shown in Figure 1. However, reality shows that the reaction to these events is not quick enough. For instance, on 1 April 2020, the Russian telco services provider Rostelcom was involved in an incident impacting more than 8800 prefixes [15]. This event influenced companies such as Amazon and Akamai. In July 2020, the ATLDC Tulix Systems caused an incident where they announced 145 incorrect prefixes. These erroneous prefixes lasted for almost an hour [16]. For example, a significant BGP leak occurred in June 2019; the provider China Telecom accepted more than forty-thousand routes from SafeHost [17]. The announcement contained prefixes already present in the global BGP table or non-existent prefixes. As a result, the China Telecom spread them to its neighbours, which caused a global issue.

### 2.2. RPKI

RPKI is a hierarchical PKI that is dedicated to securing Internet resources such as AS numbers or IP addresses. It binds the AS number and IP address to a public key via a certificate. A holder of the associated private key can perform attestation about these resources, so-called Route Origin Authorisation (ROA). ROA authorises an announced IP prefix if it is signed by the private key of the certificate covering the given IP range. Each of the five Regional Internet Registries (RIRs) holds its own RPKI trust anchor, which is the same as the root certificate in the Public Key Infrastructure (PKI). With the corresponding private key to the trust anchor, the Regional Internet registry (RIR) can release a certificate to another member to generate ROA [18,19]. RPKI contains certificates, ROAs, manifests and certificate revocation lists published in RPKI repositories and validation software (Relying Party). The validation software obtains data from RPKI repositories and validates them. As a result, it creates a set of valid ROAs.

The main drawbacks mentioned in [20] against the deployment of RPKI are:the deployment is too slow;the centralised authority, RPKI is able to revoke or change any certificate that it has issued;misconfiguration or dishonest behaviour of RPKI can damage ROA, which causes that prefix to be unavailable.

### 2.3. Border Gateway Protocol Security (BGPSec)

BGPsec is a mechanism to ensure that an AS inserts only the correct AS number into the AS path in the announced update. Indeed, the announced path correctly matches the real AS path used for the traffic forwarding. Meanwhile, BGPsec relies on the RPKI service, which provides a certificate for each AS to sign and verify records. After an AS received an update message, the AS verifies the signature, inserts an AS number of the next hop and signs the new update message. In a real scenario, the BGPsec routers can use the RPKI cache server to eliminate the overload caused by the verification process. The RPKI cache server verifies origin, validates the path in the update message and then distributes the result to all BGP routers within the AS [21]. Although BGPsec is designed to improve inter-domain routing security, there are still vulnerabilities, such as a wormhole and mole attacks [22].

## 3. Overview of Blockchain

Blockchain is a system based on a shared book across a network. It is often described as a decentralised ledger because every participant in the network maintains a replica of the ledger. There is no third party trusted authority to control the network or mediate transactions. It is the peer-to-peer system where all participants in the network collaborate in its maintenance. Decentralisation and collaboration are two pillars of this technology.Another important blockchain concept is the smart contract, a self-executed code to enforce an agreement between parties involved in the transaction without a third party. However, the contract code hides some vulnerabilities we should avoid [23].

Transactions can only be suggested by participants in the network and are inserted into the pool of waiting transactions. The next step is the transaction approval process called consensus. The algorithm of this process is determined based on the selected consensus mechanism. If the transaction is approved, it is included in the new block added to the chain. The main aim is to validate transactions and to obtain a uniform view of the chain among participants. Moreover, the reliability and consistency of the transactions are guaranteed. Every block refers to the previous block in the chain except for the genesis block. This reference is the value that is calculated from the previous block by the cryptographic function. The genesis block determines the beginning of the chain and is created by the founder of the chain. After the transaction is added to a block in the chain, it will be almost impossible to modify the transaction. A block is characterised by a hash value and timestamp, which offer possibilities to audit transactions. If an attacker wanted to change the transaction, he would have to calculate the new value for every subsequent block in the chain. For the final confirmation of a new block, we have to wait to commit several following blocks. It reduces the probability of a successful attempt to change the transaction by an attacker. Due to the immutability of records, blockchain is sometimes described as a system of proof. Despite it, if the number of nodes is low in the network, a group of malicious nodes can obtain a majority and force the rest of the nodes to accept their branch of the chain.

The consensus mechanism provides synchronised transactions across the network. Moreover, it keeps the order of performing transactions. This mechanism removes some problems of traditional distributed databases, such as inconsistent writing to the database. The disadvantage of this approach is that consensus mechanism requires much more time to complete the write operation.

Users can participate in the transaction without knowing each other. They must own the public key to derive an address for communication with others in the network and the private key to sign transactions. This mechanism achieves the anonymity of users. However, prior studies note that blockchain cannot guarantee it perfectly since transaction information for each public key is publicly accessible [24]. If the transaction has been accepted, the record of it exists in transaction history, which ensures no loss.

### 3.1. Type of Networks

Generally, we can classify blockchain into two basic types: public and private. They have very little in common except for the concept that both provide a shared ledger across the network. In a business environment, a modification of these basic types are used very often; we call these consortium or hybrid blockchains. They merge the advantages of both to achieve better performance, scalability and transparency. Table 1 summarises the benefits and drawbacks.

A public blockchain is also referred to as permissionless because anyone can join or leave the network without verifying identity and asking for permissions. The network participant can read transactions, propose new trades and expect to see them included if they are valid, and collaborate in achieving consensus. Cryptocurrencies, Bitcoin and Ethereum are the most familiar environments for using a public blockchain.

A private blockchain is also called a permissioned one. There is a central authority to control the network and manage access rules. It can require identity verification and a proof of membership from the participant. All participants are known and trusted by each other. The consensus is achieved faster because of the lower number of validators. On the other hand, the central authority can influence the rules of the blockchain, revert transactions, etc. The Hyperledger Fabric [25] can be used to establish one.

A consortium blockchain is a semi-decentralised network. In contrast to the previous case, the consensus mechanism is managed by a set of nodes from various organisations. It is mainly used by the financial sector, government organisations, etc. R3 Corda [26] and Hyperledger Fabric [25] are known as consortium blockchain.

A hybrid blockchain has several benefits, as it merges the security of a public blockchain and the efficiency of consensus of the private blockchain. Therefore, with a hybrid blockchain, a public blockchain makes data accessible to all, and a private blockchain modifies a ledger in the background. Dragonchain [27] or Orbs [28] are the most popular platforms based on the hybrid.

### 3.2. Consensus Algorithms

In this section, we review notable consensus algorithms. They are categorised into two groups: proof-based and vote-based algorithms [29]. Proof-based algorithms require that the participant performs physical work on their hardware. If it provides sufficient proof, it will obtain the opportunity to add the block to the blockchain and gains a reward. In this group, there are algorithms such as Proof of Work (PoW) and Proof of Stake (PoS). On the other hand, the vote-based algorithms prefer an election system. The participants exchange messages among each other until somebody obtains the majority of votes, for example, Practical Byzantine Fault Tolerance (PBFT). Afterwards, the election winner can add the block to the blockchain. Moreover, this type of consensus is often used in private or consortium networks in which the decentralisation degree is lower like in public networks.

Overall, the consensus algorithms are based on Byzantine Generals’ Problem [30], where the group of untrusted nodes takes effort to reach a common agreement. Moreover, the vote-based algorithms are categorised into Crash Fault Tolerance (CFT) and Byzantine Fault Tolerance (BFT). The critical difference is in the type of accepted fail; CFT algorithms can only tolerate crashed nodes. Therefore, there should be at least N/2 + 1 nodes alive. In contrast, BFT requires a minimal 2N/3 + 1 nodes, which can not be crashed or be malicious [31]. It follows that reliability is a critical feature of these algorithms, which indicates how many percentages of nodes can behave abnormally, and the consensus will not be corrupted.

Furthermore, the consensus algorithm must ensure delivering a message from the source node to the destination node. Furthermore, it must prevent overwriting or corrupting the last valid state. This feature is crucial for maintaining a uniform state among nodes. Finally, there are also properties such as scalability and performance that are contradictory. If we want to increase performance, scalability will be lower and vice versa [32]. For example, Bitcoin has high scalability with a large number of nodes in the network, but it has low performance. We can only add a new block every 10 min, and the throughput is 7 tx/s [33]. On the other hand, platforms based on PBFT reach high performance, even 10,000 tx/s [34]. However, this throughput is only in a network with a low number of nodes such as private or consortium. Otherwise, the communication overhead is increased because nodes have to know each other and exchange messages. Table 2 shows the comparison of the consensus algorithms.

Proof of Work (PoW) is the first and most widely implemented consensus algorithm. The PoW is energy-intensive because finding solution consumes a lot of energy with miners competing.

Proof of Stake (PoS) is generally more energy-efficient than PoW [43]. The miner is chosen based on their stake, with a larger stake resulting in a higher probability of adding next block [44]. Miner selection randomisation is used to prevent network takeover, and if fraud is detected, its executors lose their stakes. The PoS faces security threats, such as nothing at stake or long-range [45]. Currently, Ethereum is in the process of switching to PoS in version 2.0 [46].

Delegated Proof of Stake (DPoS) is an algorithm in which the currency holders elect a group of witnesses to exercise their powers on their behalf. The elected witnesses create blocks and validate transactions. However, they must surrender their coins to a time-locked security account to prove their credibility. In the case of malicious behaviour, they lose their stakes [47]. Moreover, some experiments try to design an algorithm with a lower level of centralisation and the same level of effectiveness [48].

Practical Byzantine Fault Tolerance (PBFT) introduces state machine replication that can run even when some the participants are malicious. The participants are sequentially ordered: one of them is the leader node, and others are known as backup nodes. The validation of the client request is achieved in three phases. However, only up to a third of the participants can be malicious for them to not have an impact on the consensus [49]. The main problem of PBFT is scalability because of the rising communication complexity with new participants [50]. This mechanism is optimal for small environments, but this also increases the risk of a Sybil attack.

Proof of Authority is a modified form of DPoS where instead of coin, the validator stakes their reputation. If the validator wants to create a block, they must confirm their identity with officially issued documentation [51]. The validator identity is visible to everyone in public.

RAFT specifies one of the three states for the node at any given time: follower, candidate or leader. Time is divided into terms, and it can be considered logical time. At the beginning of the term, the node becomes the follower. If it does not receive a heartbeat message for a certain period from the leader, it will move to the candidate state and then send a RequestVote message to other nodes. If it obtains the majority of nodes’ votes, it will become the leader for the term. Otherwise, if the leader is not elected, they will continue to the following term. The consensus process is as follows: there is one leader, and other nodes are only followers. During the term, the leader receives a transaction from the client and replicates it to followers. After the leader obtains a reply from most followers, the leader announces that the transaction is committed [52]. As the performance analysis showed, this algorithm can be sensitive to network parameters, such as packet loss, network size and election timeout [53]. If the current leader becomes unavailable for several nodes in the network, these nodes will start electing a new leader, and it may cause an unwanted split of the network.

### 3.3. Attacks in Blockchain

One of the prominent features is decentralisation in the blockchain. Nevertheless, a blockchain can face various attacks on the main chain or single nodes in the network. In this section, we mention basic attacks in the blockchain. Figure 2 shows the taxonomy of blockchain attacks.

#### 3.3.1. Fork Attacks

In this type of attack, the attacker tries to replace the most trusted chain by launching an alternative chain to gain a higher reward or push through their interests. Generally, there are “Selfish mining” and “Sybil”.

Selfish mining’s main intention is to gain unfair rewards and waste the honest miners’ computing power. The malicious node does not commit the discovered block into the network, but it keeps it privately. Furthermore, it continues to mine its blocks on the private chain to obtain a longer chain than the public chain. Meanwhile, the honest nodes mine new blocks on the public chain. If the attacker gains the longer chain, he will publish the new block into the public network. The honest nodes will join the malicious node’s branch because the system’s design prefers the longest private branch [54]. The zero block scheme [55] proposes to accept each new block within the maximum time interval to protect against this attack.

In a Sybil attack [56], the attacker creates fake identities to obtain control of the network. At the same time, the same entity manages all fake identities. Consequently, the attacker can influence voting or enforce their blocks to honest nodes and prevent the spread of the honest nodes’ blocks. The detection of this attack is easier in the private network than in the public network. The principle of protection mechanisms is that the system motivates nodes to be honest and demotivates nodes by the non-profitability of this attack. Moreover, 51% attack [57] is a particular attack in a system with Proof of Work, where the attacker tries to control more than 50% hashing power. Afterwards, it could reverse transactions, branch out the main chain or perform a Double Spending attack [58]. The last such attack was recorded on Ethereum in 2020, where over 7000 transactions were recognised [59].

#### 3.3.2. Network Attacks

In the decentralised network, the nodes are placed in different geographical areas. This type of attack tries to isolate a node or group of nodes from the remaining network. This may include a few scenarios of attack.

DNS attack, the attacker manages to manipulate a DNS entry in the registry. As a result, the honest node connects to a fake website.

The BGP hijack is an unauthorised change in the AS path, where the traffic between the honest nodes in different ASes is redirected to a malicious node. In 2018 [60], the cryptocurrency website MyEthereum was a victim of this attack, losing about 152,000 dollars.

The Eclipse attack [61] occurs when the attacker gains control over incoming and outcoming traffic of the honest node. The view of the rest of the network may be filtered for the node.

The Finney attack is a variant of Double Spending in which the attacker delays releasing the block to the public network to double-spend their assets [62].

The Timejacking attack tries to confuse the internal timer of the honest node derived from the network timer. Many fake nodes send messages with a false timestamp. As a result, the honest node will not accept a block with the current timestamp from the rest of the network [63].

#### 3.3.3. Application Attacks

Nowadays, blockchain applications are built on an aspect called a smart contract. It allows running code transparently for all participants without a third party. However, it also brings a new point for attacks. We focus primarily on significant attacks related to smart contracts in the section.

A Re-entrancy attack [64] can occur when a contract makes an external call to another contract before it performs all internal state changes. The secure order of actions can help to avoid such attacks.

A DoS external call attack occurs if a conditional or loop statement depends on the result of an external call. The callee may permanently fail using throw or revert. Therefore, the caller will not be able to complete the execution [65].

An Overflow attack [66] is caused by exceeding the value of the unit type (2^256). Because of this, the value will be set to zero. This same is true for underflow; when the value is less than zero, it will be set to the maximum value.

Insufficient gas griefing [66] is a type of attack where the attacker does not directly benefit, but he prevents the transaction from being performed. This attack can be made on contracts that accept data and use the data to make a sub call another contract. Someone executes a transaction with only enough gas to execute the transaction but not enough gas to finish the sub call.

Forcibly Sending Ether to a Contract [66] is a vulnerability in a smart contract that allows sending ethers to a contract without triggering its fallback function. It is forced by the selfdestruct() function.

## 4. Methodology

The threat model of RPKI assumes that the centralised authorities are always trusted. It does not contemplate attacks on authorities or their dishonest behaviour. The central authority can abuse its power to change or revoke any certificate that it has issued. Because of this fact, there is an apparent power imbalance between authorities and their members. Overall, some architectural deficiencies of RPKI can cause specific side effects that lead to a prefix becoming unreachable [20,67,68]. Furthermore, BGPsec should resist various types of hijacks by cryptography authentication of the whole path in the BGP announcement. However, the attacker can still create routing wormholes or forwarding loops. Unless all ASes enforce BGPSec, the partial deployment of BGPSec brings almost no benefits [22]. A blockchain can solve those problems by decentralised access without changes in the existing BGP architecture.

A number of projects are utilising blockchain to improve the security of BGPs used in global Internet routing. However, most of these address only a particular aspect with a narrow focus, reviewing particular related work with only limited comparisons. We aim to provide a comprehensive survey of designs combining blockchain with BGP and summarise our research points in the following questions:

RQ1: What existing aspects of inter-domain routing can be enhanced using a blockchain?

RQ2: What are the existing blockchain-based BGP-implemented projects? What are their technical characteristics? What types of attacks can be prevented?

RQ3: How does the choice of the consensus mechanism impact the performance of the system?

RQ4: Can blockchain performance limit BGP scalability?

RQ5: Is there any effort to combat a problem called blockchain bloat?

### Materials and Methods

We examine peer-reviewed literature in scientific databases to identify relevant journal articles and conference papers. We performed a search of the following databases: IEEE Xplore, ACM DL and Google Scholar. All the searches were executed between November 2020 and July 2021 using a combination of keywords:


*blockchain; BGP; decentralised network resource management; autonomous system; distributed address management*


A total of twenty-one papers related to blockchain used in inter-domain routing emerged from these searches directly and through forward snowballing the citations of search results. We defined inclusion and exclusion criteria for including a paper in this review. Due to the novelty of the subject material and a relatively low number of projects concerned, we selected the following permissive criteria.


**Inclusion criteria (IC):**
The paper should have been published in the last five years, i.e., after 2016;The paper should clearly describe its key contribution;From the description in the paper, it must be clear that the work aims to apply blockchain technology for enhanced security of BGP protocol;If the project has several published papers, we choose the latest published work in a journal.



**Exclusion criteria (EC):**
The paper length is less than four pages;The project is only a theoretical concept.


By applying these criteria to the 21 retrieved papers discussing 13 projects, we eliminated 52% of the papers and selected 11 for further analysis. To provide the reader with a general overview, we mention every found paper in Figure 3. Certain projects [69,70] have been excluded for further analysis because they did not meet our above-mentioned IC criteria. This was because they do not present a solution to enhance security, instead they are focused on approaches to guaranteeing the quality of service in inter-domain routing. According to them, the routing infrastructure should be agile and managed by smart contracts in the future Internet.

In the review of selected projects, we endeavour to answer research questions and focus on four key attributes. First, we look at the functionality and features provided by each solution—from the novel approach, through how they utilise the blockchain, to backwards compatibility with regular RPKI and BGPsec.

Second, but not less significant, is the performance of the proposed solutions. The speed of data dissemination and network convergence is probably the most critical factor for true Internet global scalability. Performance can be a significant bottleneck for deploying blockchains in the inter-domain routing because it depends on the selection of the consensus algorithm.

Additional improvements can be accomplished by choosing the right network type for the particular use case and by optimising the amount of data stored in one transaction. These storage requirement differences are the third parameter used in the comparison.

Lastly, there are several projects that use custom blockchains, resulting in various benefits and drawbacks. In the custom blockchain, the authors modify the consensus algorithm to optimise performance or design their own blockchain. However, there are security issues and maintaining such a system is complex.

## 5. Literature Review

This section introduces a short description of the identified projects that try to improve the lack of credibility in BGP. Later, we analyse the selected papers according to the criteria in the previous chapter in detail. Before we dive into the analysis, we underline the following observation: Even though there has been a rising interest in the application of the blockchain in inter-domain routing in the last two years, the vast majority of these articles come from the academic world. There are different ways of evaluating the performance of such a system. Because of this fact, some projects do not clearly provide information on the throughput of the system or storage consumption. Furthermore, there are few papers aimed at optimising consensus mechanisms.

To help us better analyse the projects and confront them with our research questions, we can categorise literature into the following: (G1) projects introduce an approach to enhance the security of BGP through the permissionless blockchain. One of the key reasons is that it is fully decentralised, open to all and has infrastructure. Next, (G2) projects are built on a permissioned blockchain that tends to be faster and more scalable. Finally, (G3) projects show custom-designed blockchain. The authors chose this approach because the existing platforms did not meet the specifications in G1 and G2. A summary division is shown in Table 3.

### 5.1. Permissionless Blockchain to Enhance BGP Security (G1)

In particular, (G1) projects store ROA entries into the public ledger to reach a consistent view of the global routing table. Besides verifying the origin of the IP prefix, they manage the allocation of Internet resources through a set of smart contracts. We conclude that they aim to provide high transparency with a low implementation cost.

In the design of BGPCoin [71], the authors proposed managing Internet resources as AS numbers and IP space. They tried to remove several defects of RPKI, for example, misbehaving authorities. This origin authentication framework is based on blockchain to ensure distributed and tamper-resistant management. The system is controlled by smart contracts that allow registering, allocating, assigning, updating and revoking resources. Moreover, there is a function for the aggregation of IP addresses to effectively store entries in the blockchain. Five states describe the life cycle of IP addresses in the smart contract. It makes it possible to monitor trading operations among owners and lessees. Every organisation, Regional Internet Registry (RIR) or Local Internet Registry (LIR) that holds any resources maintains a client to interact with a blockchain to perform operations on its resource. In this way, an IP prefix can be transferred from RIR to LIR and then leased to some ISP. Finally, the IP prefix binds with the AS number, and it adds the ROA record into the blockchain. The authentication of origin resources is performed by a cache-client that is installed in every AS. It learns the ROA records from the blockchain and allows the border router to request ROA from it. In addition, there is path-end-authentication, which is an alternative to BGPSec.

InBlock [72] introduces distributed autonomous organisation to the decentralised management of Internet resources. Consequently, it provides an alternative trust model to the hierarchical model of RPKI. A set of smart contracts ensure that the operations needed are performed for resource management. Moreover, the authors present security analysis and compare different types of adverse action against the RPKI and InBlock. Unlike the other mentioned proposals, except for InBlock, IANA/RIRs do not make initial allocations of IP blocks to a particular LIR. It only delegates the IP pool to InBlock, but the assignment to LIRs is performed decentralised. The current resolve is available for IPv6 address spaces, but the authors could easily extend it to AS number or IPv4 address spaces. Moreover, InBlock can only manage a subset of the address space, where decentralisation is required. Because of this fact, it does not aim to replace RPKI. The allocation record contains the allocated prefix, the holder’s account address, and the expiration date. In addition, it can include a link to external storage where the routing policies for prefixes are stored. Due to the reduced cost of transaction and blockchain size, the authors decided on this resolve. If LIR wants to obtain a prefix from the InBlock, it must first pay the allocation fee. The allocation fee is calculated in FIAT currency, whereas we have to change it into Ether. Therefore, LIR issues a transaction that contacts a third-party service to retrieve the current exchange rate between these currencies. Finally, LIR issues a transaction that computes the allocation fee in Ether. The allocation fee must be paid in 24 h. Moreover, InBlock supports aggregable allocations, which means if someone has been allocated prefixes and will ask for the next prefixes, all these prefixes can be aggregated into a large prefix. The authors evaluate several experiments targeted at cost and delay transactions. In the local environment, they performed experiments to obtain consumption GAS for every transaction. In the Mainet network, they measured the time it took to write a transaction in the blockchain and the time it took to confirm a transaction. In the paper, this concept is extended to the credibility model that allows managing access to execute transactions with Internet resources.

In [73], the authors proposed an approach where the administrator firstly uploads the IP prefixes to the blockchain. If IP prefixes comply with ROA entries, the border router of AS will download the current IP prefixes from the blockchain and announce them to neighbours. Each AS administrates its SC that RIR/LIR owns. As a result, it can prevent the propagation of false prefixes when the misconfiguration is stored in the first router in the path. This work, unfortunately, contains no transaction to register the AS number or revoke the allocated resources.

### 5.2. Permissioned Blockchain to Enhance Security BGP (G2)

In (G2) projects, ROA records are saved to a private ledger where user access can be limited. Only approved users can contribute to the consensus process. This approach’s significant advantage is high scalability and security because participants know each other. The cost of implementation is higher because we must build our own infrastructure.

ISRchain [74] is an inter-domain secure routing framework using blockchain, which prevents IP hijacking, AS-path forgery and route leak. It maintains a decentralised and consistent global view of existing ASes and IP prefix owners. The fundamental property is to validate information during routing for the ROA, AS path and route policy. Moreover, this solution aims for effective and necessary changes in the blockchain rather than all BGP announcements.As a result, it reaches better performance, decreases the number of transactions and reduces storage size. There are two types of smart contracts: Internet Resource Management contract and AS Information contract. The first contract ensures the allocation of Internet resources to ASes. Subsequently, other ASes can check the origin of the propagated IP prefixes. The second contract maintains information about a local AS related to its neighbours and business policy. Then, when the AS receives the BGP update message, it can start path validation by retrieving the ASI contracts that correspond to every AS in the path. Finally, the authors demonstrate the capability of ISRchain on real BGP incidents.

BRVM [75] is a routing verification model to resolve events that violate the shortest AS path policy. The routing promises can be easily violated between ASes while inter-domain protocols cannot to detect such behaviour. For example, AS1 makes a promise that it will always announce the shortest route as the best route to adjacent ASes. AS1 may receive multiple routes to a prefix and then decide to break the promise. Finally, it will not announce the shortest path, while AS2 cannot notice it. The main idea of BRVM is to detect two attack scenarios: the single-point attack when the promise between ASes is broken and the multi-point collusion attack when several ASes behave dishonestly.

BlockJack [76] is a system based on a consortium blockchain to block BGP hijacking attacks. The system is composed of three modules: Blockchain, Profiler and Dispatcher. The Profiler provides an interface to communicate between the blockchain network and the Dispatcher module. The Dispatcher module captures any new prefix in the BGP table in the router. If the prefix is of internal AS origin, a request is sent to the Profiler to add the prefix to the blockchain. This function is called prefix authorisation. Another process of Prefix Verification checks whether the prefixes received by the router from its neighbours fit the data in the blockchain. In case the prefix is marked as the source of a potential collision in the routing table, the Dispatcher sends filter commands to the router to restrict an incoming announcement from the AS origin of the inspected prefix. The authors evaluate the time processing for handling prefix authorisation and prefix verification.

DRRS-BC [77] offers a registration framework in the inter-domain routing to protect the origin IP prefix by introducing blockchain. It establishes a global ledger that saves IP prefixes and AS numbers between multiple organisations and ASes. It perfectly solves the security problems of the centralised authentication in traditional BGP. Security analysis proves the resistance to prefix and subprefix hijacking attacks. Moreover, the results show the scalability of the system and the impact of the block size on the processing efficiency of the system.

### 5.3. Custom Blockchain to Enhance Security BGP (G3)

The last type of approach (G3) is mainly aimed to improve scalability. Some of the investigated projects use the technique that is called sharding. It divides the nodes into subblockchains or shards, where they store the history of changes and process the transactions of the shard to which they belong. In most cases, these projects are increasingly closer with the update to Ethereum 2.0 [46].

In [78], the authors propose a blockchain-based system that records operations related to IP prefix allocation and validates paths towards these prefixes. This system can work in parallel with existing systems, such as RPKI. The transactions are divided into two groups: IP allocation transactions and BGP path transactions. The first group follows business relationships between Internet Registry (IR) and AS or AS and AS. Moreover, it also stores the lease duration for the current holder of an IP prefix. The BGP path transactions follow if the whole path is valid from the source AS to the destination AS. However, they do not maintain end-to-end AS paths because the prefix is propagated in sequence, and every announcement of the new prefix is saved into the block. The whole path is validated based on the performed transactions in the sequence from source to destination.

RouteChain [79] aims for swift consensus among ASes to accelerate the validation process. Therefore, they design a bi-hierarchical blockchain in order to improve performance and minimise the delay. To achieve this, they decided to use a consensus protocol Clique that belongs to Proof of Authority protocols. ASes are divided into subgroups according to geographical proximity. The authors calculated that for the year 2018, we need to divide 88 721 existing ASes into 298 subgroups to achieve the minimum number of messages necessary to confirm a transaction. The distribution of ASes can be influenced by policy or other relationships among ASes in the real world, which can cause worse performance and may not be as close to the ideal situation. The partial hijacking attack may be neutralised by consensus in a subgroup if an attacker and victim AS are members. On the contrary, a consensus among subgroups is needed to detect a complete hijacking attack.

The paper of [80] presents a novel way to allocate and delegate Internet resources using blockchain. The prototype is built on a modified PoS algorithm. The validator is selected according to how many IP prefixes he holds for the next block. By chaining different transactions, they can replicate the operation hierarchy of the RPKI. Moreover, the authors argue there are several advantages making PoS suitable for managing IP prefixes. With the same intensity, they pay attention to the algorithm’s resistance against creating monopolies in PoS. To eliminate this risk, they modified the PoS algorithm and added a group signature to the message. The experimental section investigates the throughput and the block time depending on the Boneh–Lynn–Shacham signature. Finally, they estimate storage capacity requirements.

While the other solutions use one of the existing consensus mechanisms, ROAchain [81] offers its own optimised consensus algorithm. This algorithm aims to protect from vulnerabilities that are typical for PoW or PoS, such as Sybil attack, forking, etc. Moreover, they decided to ensure better scalability and throughput of the algorithm to minimise the impact of the blockchain on the BGP. To cope with this challenge, the verification and consensus process is split into shards for parallel processing. However, the allocation of Internet resources, validation of the whole AS path and route policy check are absent. It focuses on the proposal for the novel consensus algorithm with improved security and performance than traditional algorithms. On the other hand, it may be difficult to maintain this algorithm without community support.

## 6. Origin Validation

As we explained in Section 2, BGP assumes all the participants are honest in the network. Because of this fact, the receiver verifies neither the origin nor integrity of the BGP announcement. Due to the raised number of successful attacks against BGP, the RPKI document was launched in 2012. The main aim of the certificate chain is to prove the origin of the announced prefix. An RPKI certificate holder can sign a Route Origin Authorisation (ROA) entry that binds the IP prefix with a given AS. Based on this, the AS is entitled to advertise IP prefix to the rest of the network. At the same time, ROA is inserted into the RPKI repository, where a third party can validate it.

Most of the analysed resources point out two main problems. Firstly, the deployment of RPKI is too slow. According to the global RPKI deployment monitor [82], the global routing table contains more than 68% of unique Prefix/Origin pairs that RPKI cannot validate. One possible reason can be the second problem of RPKI, the centralisation. In the RPKI model, the higher-level entities can modify issued allocations by low-level entities. Indeed, this opens the door to several problems, such as the higher entity abusing power or a single point of failure if a third party attacks the top entity. This can have an impact on the routing in the whole network. Nevertheless, we or the authors of papers do not claim that the above-mentioned RPKI entities abuse their power, we only point out a weakness of the centralised architecture in RPKI. We summarise analysed projects in Table 4.

For the reasons highlighted above, the analysed papers proposed validating ROA using blockchain. The ROA entries are saved to the distributed ledger as transactions that cannot be modified, and network participants can quickly validate them. Moreover, in G1 papers, [74,80], the authors focus even on the distributed management of Internet resources. They make an effort to hold the sequence of operations as in the hierarchy of RPKI.

BGPCoin [71] is built on a set of four SCs that IANA may deploy in the blockchain. Each of these SCs provides an interface for another service. The first SC registers part of the IP space and allocates it to RIR/LIR. Likewise, RIR can register AS numbers. The second SC interface is determined for RIR/LIR to allocate IP prefixes and AS numbers to its subLIR or ISPs. The IP prefix holder publishes an ROA record that binds the IP prefix to the AS number through the third SC. Due to the effective transport and the minimum transaction cost, it implements the aggregated ROA mode and the minimal ROA mode. The last SC validates ROA records but also allows an auditing mechanism to address anomalies. Finally, each AS has a cache client that synchronises ROA records with blockchain, which decreases the time response and the number of requests in the blockchain.

In this work [72], the authors point out the autonomous allocation of IP prefixes. Firstly, the manager deploys SC in the Ethereum network, and then he activates it through a transaction that determines the address pool available and the address allocation fees. Later, the SC does not allow modifying these parameters. Similarly, the manager cannot revoke existing allocations and prevent any party from performing a transaction to obtain a new prefix or renew an existing one. The prefix allocation process is executed in two steps. In the first phase, LIR requires computing the allocation fee in ETHER through executing two transactions. The first one obtains the exchange from InBlock. For this purpose, InBlock uses services from a third party. Once the exchange rate is fetched, the second transaction determines the allocation fee. If LIR does not make a payment within 24 h, the fee becomes invalid. Then, LIR requests to obtain a prefix by selecting one of two different types of transactions. The first transaction is used when LIR does not hold any prefixes. On the contrary, the second transaction allocates a new prefix considering previous allocations. If the prefix is assigned to LIR and the aggregated prefix is available, the transaction is successful. Finally, the prefix holder can assign a more specific prefix to its customer or register an ROA record that allows for the propagation of the prefix in a BGP advertisement.

In ISRchain [74], the authors present two types of SCs: IRMContract and ASIContract. IRMContract is owned by IANA, responsible for the registration and revocation of IP prefixes and AS numbers in the blockchain. IRMContract administrators, RIR or LIR, execute the allocation, delegation of IP prefixes and ROA record creation. When IANA or IRM registers the unique AS number, it will create an ASIContract. At the same time, IRMContract will save a mapping between the AS number and its ASIContract address. Every AS administers its ASIContract to store a list of AS peers and business policies. In this way, each participant AS in ISRchain can validate the compliance with the business policy in the received BGP announcement. Due to the effective management, IRSConract stores only a mapping between AS number and IP prefix. The IP prefix, AS number and period are input parameters for register/allocate/revoke prefix transactions.

IPChain [80] brings only operations of the allocation and the delegation of IP prefixes to the new holder. In the delegation operation case, the new holder cannot further allocate IP prefixes to other customers. However, the registration of IP prefixes and AS numbers is absent. All the address space is assigned to IANA in the genesis block to allocate IP blocks to RIR/LIR. Then, they may allocate or delegate IP prefixes to their customers. The final holder of the IP prefix creates an ROA record. The authors do not mention a formal definition of transactions and the set of used SCs.

A different approach is taken in category G2 and G3, where the authors focus only on keeping a consistent global view, except [74,80]. However, they do not replicate the hierarchy structure of operations from RPKI in a set of smart contracts. Likewise, the process of assigning an AS number to ISP or end-user absents. The main aim is to guarantee the ROA repository’s consistency in each AS and provide an effective consensus algorithm.

For example, in [78], the allocation and delegation operation are performed by a single transaction that assigns an IP prefix from the current AS holder to the new AS. The transaction contains input parameters, such as source lease, lease duration for the current owner of the prefix and lease duration for the new owner. If the lease duration is greater than the source lease, the transaction will fail. The transfer flag determines that the new owner can transfer the prefix to another AS.

RouteChain [79] implements the division of ASes into subgroups according to geographical proximity. Each subgroup shares a single ledger with AS paths of its members. An AS presents its prefixes in the transaction to all subgroups. If subgroups approve the transaction, the affected subgroup updates its routing table, and the transaction result is saved in the global chain, which is shared among subgroups. The authors argue that this approach makes consensus more effective and decreases the propagation delay.

In [81], the ROA transaction contains an IP prefix and an AS number, including expiration time. Overall, it describes only three operations: register, update and revocation prefixes.

BlockJack [76] stores ROA entries into a consortium blockchain. The system consists of two main functions, namely prefix authorisation and prefix verification. After the IP prefix owner successfully authorises the prefix, it is recorded in the local ROA cache. If AS successfully verifies the received announcement, then it is registered into the local ROV cache. This approach minimises communication with the blockchain. They assume that if the BGP update interval is 30 s, BlockJack can authorise a maximum of thirteen prefixes per interval, with an average authorisation time of 2.16 s. In the same interval, it can verify more than 300 prefixes, with an average verification time of 0.09 s.

DRRS-BC [77] maps four kinds of functions from real-world IP address management to transactions, such as register, lease, assign and revoke.

## 7. Path Validation

To validate the whole path, we have to investigate every hop in the path. In the previous section, we discuss several methods to validate the origin of the prefix. As a next step, we focus on the remaining part of the path.

BGPCoin [71] supports path-end authentication by adding a list of allowed adjacent ASes for every AS in client SC. A holder prefix can modify the list of its ASes by sending the update or delete transaction. The cache periodically synchronises the allowed list of origin ASes in received announcements and sends the results to BGP routers. If the end hop AS before the origin AS in the advertised path does not exist in the allowed list of origin ASes, the BGP router will reject the announcement. This solution only eliminates attacks aimed at the last hop in the path where the attacker can claim to be directly connected to the origin AS.

In the design of [78], the authors developed end-to-end path validation. The basic principle is that they investigate learned announcements in sequence. Once the system propagates the learned IP prefix from its neighbours, it must publish transactions with IP prefix attributes. The published transaction contains the following attributes: the prefix to be propagated, the signature of AS that initialises the transaction, the list of ASes from which the prefix was learned and the list of destination ASes. As a result, it brings a set of partial paths towards the prefix that are reliable. Moreover, the output is transformed into directed AS-level graphs per prefix. Based on the topology associated with the prefix, the node can verify if the originator of the announcement has a valid path to the prefix.

In Routchain [79], every BGP announcement is considered as a transaction to ensure a consistent global view of all AS paths. As mentioned in V, this solution uses the sharding approach to accelerate the process of approving a new transaction. The blockchain’s data structure for each subgroup keeps track of the routing paths of all ASes within the subgroup. However, this design may limit the implementation of policies between ASes in the subgroup. Thanks to the analysis of partial and complete attacks in the paper, we assume that they are able to validate every hop in the path even though they do not mention the process of path-end or end-to-end path validation.

ISRchain [74] introduces the PathValidation algorithm to validate a path. In Section 5, we describe the ISRchain proposal that uses IRMContrat and ASIcontrat, which is especially relevant for this part. Every AS obtains ASIcontract after it registers the AS number. This SC refers to local information about AS as status and peers. The AS owner updates these data daily. The BGP update receiver will initiate path validation by identifying contracts corresponding to ASes in the path as soon as he retrieves the ISRcontract in the blockchain. Then, he investigates the contracts of ASes in the sequence. Upon verifying the existence and state of the audited AS, he checks whether announced neighbours of the audited AS are its real immediate neighbours. Finally, if the validation of each peer in the path is successful, the router will accept the message.

## 8. Policy Validation

Among providers exist various types of business agreements that determine import and export policy of routing. In the simplest scenario, we know business relationships as follows [83]:customer-to-provider, where an AS customer obtains connectivity from an AS provider for payment;peer-to-peer, where the two ASes exchange data without payments.

The business policy used for export routes claims that customer routes may be propagated to all neighbours, and peer or provider routes can only be further propagated to customers. Most of the analysed projects do not deal with these policies except for [74,75]. Indeed, the violation of route policy can cause a route leak attack.

In the section above, we mentioned that every AS in the ISRchain system has an ASIcontract, which keeps the information about local ASes. One of the items in this SC is peerings, where every pair maintains a flag that marks whether its neighbour AS is or is not a provider. If an AS receives an announcement, it will investigate each hop in the AS path to decide whether the route policy was satisfied. Ultimately, the authors provide formal proof for three scenarios of behaviour:AS receives a route from its customer;AS receives a route from its peer;AS receives a route from its provider.

In BRVM [75], an AS propagates a route to an adjacent AS and constructs a route proof with its signature. Moreover, it includes the class of route that represents the length of the AS path and the pointer to the previous root proof. Then, it is sent to the routing verification system, which must verify it and insert it into the blockchain after achieving consensus among the verification nodes. Thanks to the route proof chain, the route receiver can investigate whether the agreements are enforced among ASes and whether it has received the route with the shortest available path. To evaluate the performance, they conducted experiments in a hierarchical and linear topology. Similarly, they examine the use of local cache to accelerate the verification process.

## 9. Scalability

Scalability is one of the significant problems in the blockchain network. The key metrics that apply to the evaluation of scalability in blockchain are latency and throughput. Latency is the time taken to generate the next block. In other words, it is the time from the point that a transaction is submitted to the point that it is confirmed. Throughput is the rate at which blockchain completes transactions in a defined period. This rate is considered as transactions per second (TPS). Overall, if we want to provide a large amount of security, the capability to handle more transactions will be limited. In this section, we will critically discuss the scalability of analysed papers. We summarise the scalability in Table 5.

In order to evaluate the scalability of analysed papers in the real inter-domain scenario, we must obtain BGP traffic data according to the latest measurements. By January 2022, the routing table contained over 906,000 prefixes. The size of the routing table rises every year, but the amount of new records in 2021 was slightly lower compared to the previous two years. Throughput decreased on average to 50,000 updates per day in the last year. For the 14-day BGP profile from 3 January 2022 to 17 January 2022, the average number of prefix updates per second was 12.08. On the other hand, the peak prefix update rate reached over 8494 prefixes per second. The network topology remained almost consistent so that the average AS path length decreased slightly under 5.5 for this period. Thanks to the rise of approximately 135 additional routing entries per day, it is predicted that the routing table will contain over 1,133,000 entries by 2027 [84,85,86].

One of the main requirements for a consensus mechanism is the throughput. To collaborate with the existing BGP system, a blockchain-based solution must reach the same or higher level of throughput. Otherwise, the processing of update messages will not be effective enough to prevent the abuse of IP prefixes.

The projects [71,72,73] are based on the public network Ethereum using a PoW algorithm that offers high scalability, but its lower throughput would limit requisites of existing BGP. For our transaction to be approved as soon as possible, we must compete with other participants in the network with a higher gas price. Without the guarantee of the throughput, we would bring a certain degree of instability to BGP. Moreover, this solution would consume too much energy to contribute to a healthy planet. On the other hand, we could obtain a fully decentralised system to control Internet resources.

Other projects [74,75] are built on an open-source blockchain platform with a private chain. Consensus mechanisms that are used in these platforms reach high throughput but with lower scalability. If we assume that each AS participates in the consensus process, some consensus algorithms may be ineffective. In PBFT, the drawback is the exponentially increasing message count if we add a new node. For example, when BGP currently has over 72,800 ASes [84], this gives us a minimum of approximately 4.23 × 10^10^ [87] messages for one request. As a result, there is a lot of message overhead.

Authors of multiple projects modified an existing consensus algorithm and then deployed it to their platform. The method of shading is used to increase scalability. The key idea is to divide the network into subnetworks, called shards. Subsequently, each shard will process a different set of transactions. The primary aim is to allow many more transactions to happen in parallel at the same time. Indeed, there are security issues to protect shards against attacks, such as a single shard takeover attack. Therefore, each shard has to satisfy the byzantine validator limit. This limit is 33% average of validators which can be malicious. In the case of the BGP environment, there are two approaches to split ASes into shards. Ref. [79]’s design divided ASes based on their geographical proximity, but this approach can be limited by the AS policy. On the contrary, ref. [81] uses randomly assigned ASes to different shards.

The throughput capacity of most of the analysed projects reaches a borderline value of the existing BGP system except for [75,77,81]. In these two projects, we need to perform further measurements to confirm their results that are evidently better than in other projects. Regarding the peak prefix update rate and increasing the count of updated prefixes per day, throughput will have to be increased. At the same time, the security of blockchain cannot be weakened. A summary of throughput is shown in Figure 4.

Another critical parameter is the latency. According to BGP Updates [85], the average convergence time per day takes 50 s. To prevent an attack, the duration of the deployment of a new block has to run less than this interval. Most of the analysed projects reach lower latency than convergence time.

One of the most challenging problems currently is blockchain bloating. The node has to run as a full node if it wants to validate transactions. In this case, the node stores the full data set in the blockchain history. The size of the chain is growing every year. We will try to estimate how much the size of the chain increases for 2020 in analysed projects. We cannot estimate all projects because some projects do not mention any information about the size of the transaction or used GAS. A GAS unit is the smallest type of work that a miner performs to add a transaction to a block. In projects that provide the GAS used for the transaction, we can approximately determine the size of the transaction (in bytes), as expressed by
$$tx\text{\_}size=\frac{used\text{\_}gas}{avg\text{\_}gas\text{\_}limit}\times avg\text{\_}block\text{\_}size$$

The size of the chain can be expressed by
chain_size=365×avg_updated_prefixes_per_day×tx_size

Figure 5 represents the storage size comparison of projects which mention the size of transactions.

## 10. Discussion

We summarise the following key findings in regards to the research questions based on the analysis of projects concerning blockchain assets for inter-domain environments presented in the previous section.

*RQ1:* From the projects we selected for our review, we conclude that the following aspects of inter-domain routing can be improved:The centralisation can be replaced by a decentralised approach to manage Internet resources. All providers and registrars can collaborate on the assignment and allocation of prefixes and AS numbers. It prevents the abuse of position or the adverse effects caused by attacks against the central authority.The blockchain can provide immutable storage to store ROA records, registrations, allocations or duration leases. We will obtain unique proof about the existence of an asset. Moreover, we can easily verify assets, and, thanks to the transaction history, we can investigate incidents.The BGP is a protocol that does not directly include security mechanisms. However, the blockchain can be used as a supporting mechanism for BGP to secure the whole AS path against attacks, such as BGP hijacking or route leak.The decentralised approach in a blockchain can ensure high availability to validate and audit resources. Given that the full node keeps a copy of the whole chain, the failure of several nodes cannot cause service unavailability or consistency violation of BGP records.Smart contracts can bring a higher level of transparency between participants in inter-domain routing. This way, the registrators and providers can easily implement agreements with each other without a third party. The smart contract will be automatically executed when the conditions are fulfilled.Most analysed projects can run in parallel with existing inter-domains systems, such as BGP, BGPSec and RPKI, without requiring any changes. Thus, they add a further level of security for inter-domain routing.

*RQ2:* Nowadays, several designed blockchain-based solutions increase security in inter-domain routing, although only a part of them have a published source code. As discussed in our analysis and shown in Table 5, many implemented projects use the Ethereum or Hyperledger platform. However, a large number of projects decided to create a consensus mechanism that would provide high scalability. We next found that analysed projects provide various levels of AS path validation. Except for [75], all projects validate the origin of the AS path. Several projects also provide the authentication of the last hop or whole AS path. Another promising finding is that there are works focused on checking the AS policy and business policy. Based on these facts, there are various levels of protection against attacks that are summarised in Table 6.

*RQ3:* The Blockchain Trilemma addresses the three critical aspects faced in creating a blockchain system that is scalable, decentralised and secure. Regardless of this fact, it is difficult for any blockchain system to achieve all aspects. As shown in Table 2 and Table 5, the consensus mechanisms allow high throughput in private platforms but with a lower level of decentralisation. On the other hand, the level of tolerance of failed nodes can limit throughput. In PBFT, the number of exchange messages among nodes exponentially increases to reach consensus after adding a new node. Therefore, if we want to keep up the throughput, we should choose a fault-tolerant algorithm. Even though the Ethereum is fully decentralised and allows good vertical scalability, the throughput is lower than ideal for inter-domain routing. Another promising finding is that the sharding approach brings high throughput, but it still faces security issues, such as a single shard takeover attack. Similarly, the important parameter is the latency which should be lower than the average routing convergence time so that false information will not spread to the whole network. We found out that part of the analysed projects achieve borderline latency values.

*RQ4:* The adoption of blockchain technology in inter-domain routing has certain limits at the current state, resulting from our analysis. The throughput of the reviewed works is approximately the same or slightly higher than average prefix updates per second. In the case of the peak, we recognise that when the existing system may process thousands of updates per second, its throughput is low.

*RQ5:* Blockchain bloat is generally a problem of any blockchain system. Especially in inter-domain routing, there are hard requirements related to storage because devices have a small size of memory. According to some above projects, we can implement a cache module in each autonomous system that will be a member of the blockchain network and handle requests to validate prefixes from border routers. Finally, we found out that some reviewed projects implement aggregated functions to compress the number of prefix records in the blockchain.

From the analysis presented in this paper, we sketched several issues that are yet to be solved. The main technical limit of the designed solutions is the scalability issue. As shown earlier, there is a requirement for high throughput at the peak, but the existing blockchain technologies cannot yet satisfy this. The BGP convergence delay is high for some real-time applications [88]. Because of this fact, the consensus must be reached in a short period. Furthermore, the importance of QoS parameters on the path rises with the increasing use of real-time applications. Novel approaches using smart contracts can help ASes to establish and maintain paths with negotiated QoS parameters. When discussing the research limitations, it is important to keep in mind that all results and conclusions of this survey are limited to the state of the art and results presented in the analysed papers with respect to the research questions provided in this paper. Furthermore, the performance comparison was only conducted using the results published in the papers, as not even half of the papers provided the implementation source code. Moreover, there are other open issues regarding security, especially if we increase throughput. Apart from these facts, the integration of a business policy should be more profound. We believe that these limitations present opportunities for further research.

## 11. Conclusions

To conclude, we can see that centralisation can be replaced by a decentralised approach to manage Internet resources. It prevents the abuse of a position or the adverse effects caused by attacking against a central authority. The throughput capacity of most of the reviewed projects reaches a borderline value of the existing BGP system. Another critical parameter is latency. Most of the reviewed projects reach lower latency than convergence time. The blockchain can provide immutable storage to store ROA records, registrations, allocations or leases duration. This way, we obtain unique proof of asset existence. Moreover, we can easily verify assets, and we can investigate incidents thanks to the transaction history. The decentralised approach in a blockchain can ensure high availability. Given that the full node keeps a copy of the whole chain, the failure of several nodes cannot cause service unavailability or violation consistency in BGP records. Smart contracts can bring a higher level of transparency between participants in inter-domain routing. This way, we can enforce business agreements or policies between providers without any third party being involved.

Most reviewed projects can run in parallel with existing inter-domains systems, such as BGP, BGPSec and RPKI, without requiring any changes. Thus, they add a further level of security for inter-domain routing. When discussing Ethereum, on the one hand, it is fully decentralised and allows good vertical scalability. On the other hand, the throughput is lower than ideal for inter-domain routing. The adoption of the current implementations of blockchain technology for the purpose of increasing inter-domain routing security has performance limitations that need to be resolved before widespread utilisation, as mentioned in our analysis. As shown earlier, there is a requirement for a high throughput at the peak, which the existing blockchain technologies cannot yet satisfy. Together with open issues regarding security, there are areas that need to be addressed by further research in this field.

In this article, we present the first comprehensive survey of blockchain-based solutions for increasing inter-domain routing security of BGP. We systematically categorise them based on the type of blockchain used. The main objective is to provide foundational knowledge for researchers interested in this area and to suggest open issues that need to be resolved. To their end, our survey includes the analysis and comparison of their capabilities, performance parameters, level of protection against BGP attacks, scalability and limitations.

## Figures and Tables

**Figure 1 sensors-22-01437-f001:**
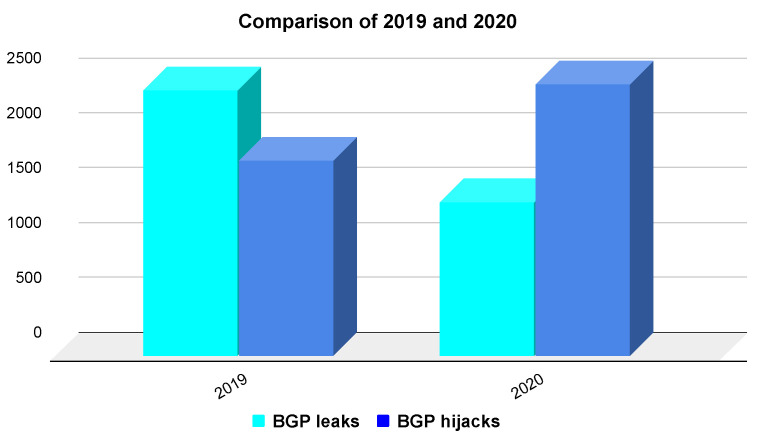
Comparison BGP events of 2019 and 2020 [13,14].

**Figure 2 sensors-22-01437-f002:**
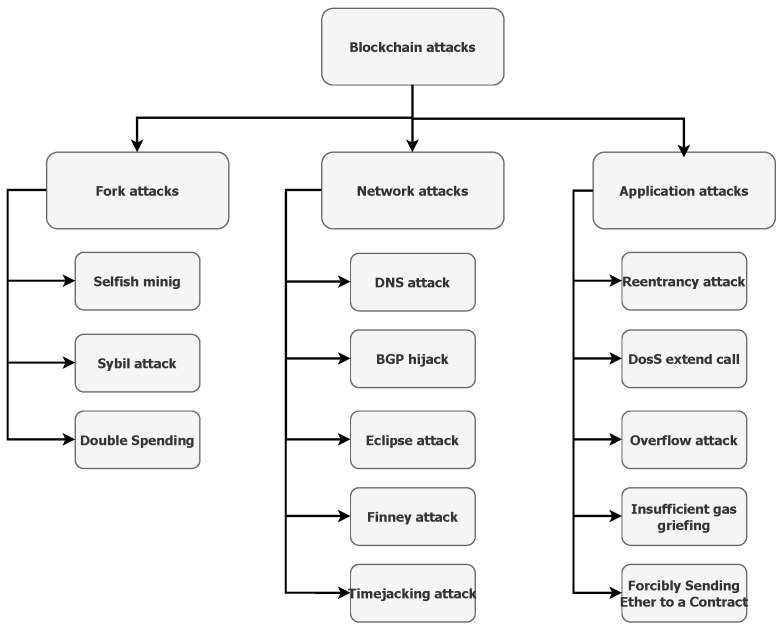
Taxonomy of blockchain attacks.

**Figure 3 sensors-22-01437-f003:**
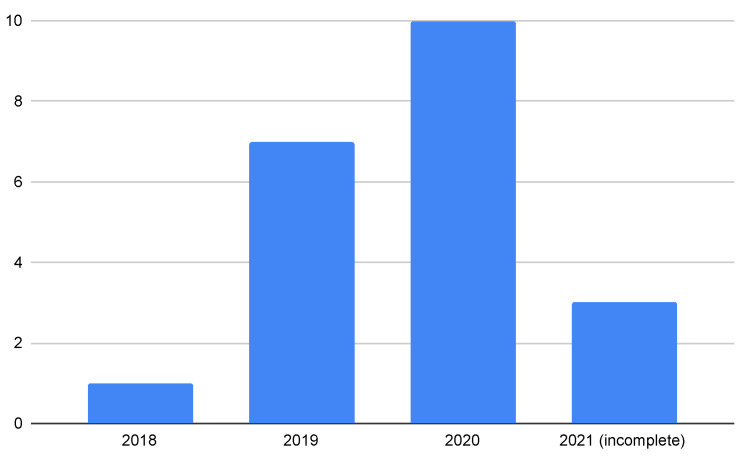
Number of published papers on blockchains for BGP per year. The plot refers to all papers of projects included in this study.

**Figure 4 sensors-22-01437-f004:**
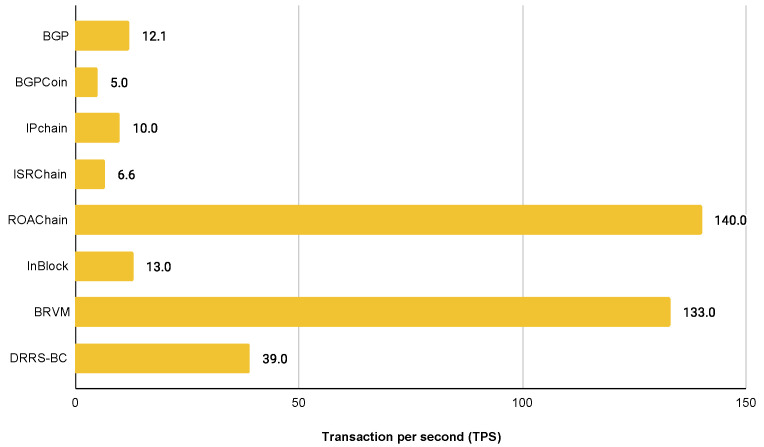
The comparison of TPS of the reviewed projects with BGP.

**Figure 5 sensors-22-01437-f005:**
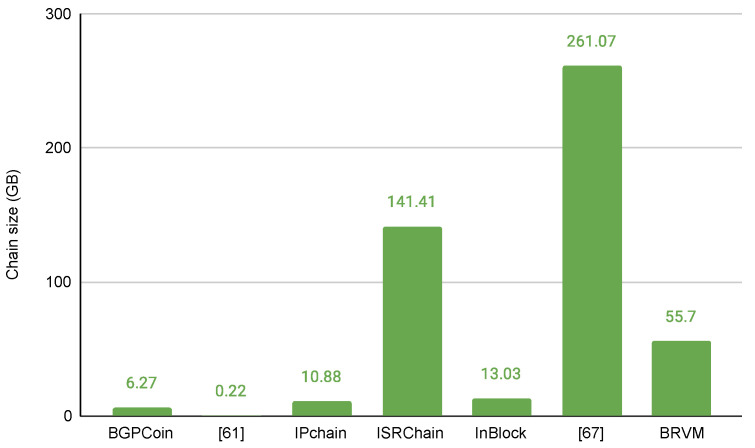
The comparison of storage size [61,67].

**Table 1 sensors-22-01437-t001:** Comparison of blockchain network.

	Public	Private	Consortium	Hybrid
Access Read	All	One organisation	Group of organisations	All
Access Write	All	One organisation	Group of organisations	Selected nodes
Centralisation	No	Yes	Partial	Partial
Credibility	High	Low	Medium	Medium
Energy Consumption	High	Low	Low	Low
Throughput	Low	High	High	High
Transparency	High	Low	Medium	High
Transaction Validation	All	Chosen participants	Chosen participants	Chosen participants

**Table 2 sensors-22-01437-t002:** Comparison of consensus mechanisms [35,36,37].

Consensus	Fault Tolerance	Throughput	Scalability	Example
PoW	<1/4 nodes	Low	Good	Bitcoin [38]
PoS	<=1/2 stakes	Medium	Good	Qtum [39]
DPoS	<=1/2 validators	Medium	Good	EOS [40]
PBFT	<1/3 nodes	High	Weak	Zilliqa [41]
RAFT	<=1/2 nodes	High	Weak	Hyperledger Fabric [25]
PoA	<=1/2 nodes	High	Good	VeCahin [42]

**Table 3 sensors-22-01437-t003:** Thematic categories in identified projects.

Group	Description	Papers
G1	Permissionless blockchain	[71,72,73]
G2	Permissioned blockchain	[74,75,76,77]
G3	Custom blockchain	[78,79,80,81]

**Table 4 sensors-22-01437-t004:** Comparison of analysed papers with inter-domain security using blockchain.

Work	Management of Resources	Validation			
		ROA	AS Path	AS Path-End	AS Policy
BGPCoin [71]	YES	YES	NO	YES	NO
[78]	NO	YES	YES	YES	NO
RouteChain [79]	NO	YES	YES	YES	NO
IPchain [80]	YES	YES	NO	NO	NO
ISRchain [74]	YES	YES	YES	YES	YES
ROAchain [81]	NO	YES	NO	NO	NO
InBlock [72]	YES	YES	NO	NO	NO
[73]	YES	YES	NO	NO	NO
BRVM [75]	NO	NO	NO	NO	YES
BlockJack [76]	NO	YES	NO	NO	NO
DRRS-BC [77]	NO	YES	NO	NO	NO

**Table 5 sensors-22-01437-t005:** Comparison of blockchain implementations in analysed papers.

Work	Type of Network	Platform	Consensus	Latency (s)	Throughput (TPS)	Average TRX Size (B)	Resource Code
BGPCoin [71]	Public	Ethereum	PoW	25	5	288	NO
[78]	Private	Python chain	PoW	54.7	-	10	YES
RouteChain [79]	Private	-	Clique	54.23	-	-	NO
IPchain [80]	Private	Pyhton chain	PoS	40	10	500	YES
ISRchain [80]	Private	Qourum	Raft	2.8	6.62	6.5K	YES
ROAchain [81]	Private	Golang chain	Sharding	73	140	-	NO
InBlock [72]	Public	Ethereum	PoW	18	13	559	YES
[73]	Public	Ethereum	PoW	54.7	-	12K	YES
BRVM [75]	Private	Hyperledger	DPoS	3	133	2.56K	NO
BlockJack [76]	Private	Hyperledger	Raft	2.16	-	-	YES
DRRS-BC [77]	Private	Hyperledger	-	9–46	39	1–5M	NO

**Table 6 sensors-22-01437-t006:** Summary of attacks prevention.

	BGP Hijacking			
Work	Origin	Forgery		Route Leak
		Path-End	Path	
BGPCoin [71]	✓	✓	✗	✗
[78]	✓	✓	✓	✗
RouteChain [79]	✓	✓	✓	✗
IPchain [80]	✓	✗	✗	✗
ISRchain [80]	✓	✓	✓	✓
ROAchain [81]	✓	✗	✗	✗
InBlock [72]	✓	✗	✗	✗
[73]	✓	✗	✗	✗
BRVM [75]	✗	✗	✗	✓
BlockJack [76]	✓	✗	✗	✗
DRRS-BC [77]	✓	✗	✗	✗

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
