# Peer review of "Secure Inter-Domain Routing Based on Blockchain: A Comprehensive Survey"

_sensors, 2022, doi:10.3390/s22041437_

Round 1
Reviewer 1 Report
The paper is a survey on blockchain solutions able to enforce BGP security. In particular, the paper explains how blockchain technology can provide an alternative to prevent the false origin of IP prefixes or hijacking autonomous system paths.
The paper sounds good and it is easy to read....
also conclusions can be certainly endorsed.....
Author Response
This reviewer did not provide any questions or comments on the content to be changed. Instead, the reviewer writes: “the paper sounds good and it is easy to read, also conclusions can be certainly endorsed.”
We are very thankful to this reviewer for his positive feedback on our work. It is always gratifying for a researcher to hear this kind of feedback.
Reviewer 2 Report
The authors should mention more references in the introduction.
They should explain RIR (Regional Internet Registry) on page 4, line 153, when it is used for the first time.
Proof of Authority consensus mechanism should be included in the third paragraph.
On line 299, please remove the dot from “Some systems. such as Peercoin”.
The numbers from the example presented on lines 853-855 should be justified eventually with some bibliography reference.
On line 929, authors should specify the number of a table where it is written “Table ??”.
Author Response
We are very thankful to this reviewer for his feedback on our work.
This reviewer states that “The authors should mention more references in the introduction.”
We added references [2], [3] and [4] in the introduction.
The review continues with a suggestion “They should explain RIR (Regional Internet Registry) on page 4, line 153, when it is used for the first time.”
We explained the acronym RIR in paragraph 2.2 .
Nevertheless, the reviewer continues by pointing out that “Proof of Authority consensus mechanism should be included in the third paragraph.”
We included the short description of Proof of Authority (PoA) consensus in the third paragraph, and we also added PoA to the comparison table “Table 2”.
The review continues with a suggestion “The numbers from the example presented on lines 853-855 should be justified eventually with some bibliography reference.”
We added the bibliography references that justify the numbers. However, we updated the count of ASes to January 2022. The minimum count of the transferred messages was calculated by the formula:
1 + 3f + 3f ( 3f - f ) + ( 3f - f + 1 )( 3f + 1 ) + ( 3f -1)
where f is the maximum number of faulty nodes.
Last but not least, also this reviewer reminds, that typos should be fixed :
“On line 299, please remove the dot from “Some systems. such as Peercoin.”
“On line 929, authors should specify the number of a table where it is written “Table ??”
We checked the manuscript and fixed multiple typos.
Reviewer 3 Report
The authors survey the use of blockchain for preventing attacks against routing information in the BGP protocol of the internet. Their survey describes those attacks and considers proposed ways to prevent them, including new attacks possible after incorporating blockchain. This is a problem of practical importance and I do not know of a similar survey. Overall, the paper is well organized and covers well its topic, including many references. This paper should be useful to researchers and practitioners of internet security.
Possible improvements:
Table 2 is far from where it is referred in the text, this is annoying for the reader.
Sections 2 and 3, background and tutorial on BGP and blockchain, can be shortened, especially the section on blockchain, which can be found in many places.
Section 5.1 and other sections that describe protocols could have shown some of them using UML sequence diagrams. Word descriptions of procedures to improve security or describing attacks are hard to follow without diagrams. It is not possible to do that for all the papers they consider but doing this in a few papers would make the survey much more readable.
Section 10. In the discussion of RQ3, after saying that Byzantine fault-tolerant consensus increases messages exponentially with new nodes, the authors indicate that this protocol should be chosen to increase throughput. This is a contradiction.
There is no section on threats to the validity of their study. It should be added.
References: Why some names are all in caps, others all in lower case?
The English is understandable but awkward, with many grammar errors. A few examples:
“The comparison shown in Fig. 1.”, should be: “… is shown in Fig. 1.”
2.2 “… the deployment is too slowly…” (line 158), should be “…the deployment is too slow…”
3 “…maintains replication of the ledger.” (line 176), should be “…maintains a replica of the ledger.”
3.2 “…scalability will lower…(line 271) should be “…scalability will be lower…”.
3.3.1 “…the attacker tries to enforce their blocks” (line 355). This is bad English, and it is not clear what it means to “enforce a block”.
Author Response
This reviewer states that “Sections 2 and 3, background and tutorial on BGP and blockchain, can be shortened, especially the section on blockchain, which can be found in many places.”
The blockchain section was shortened, especially the subsection “Consensus algorithms”. Now, there is only essential information about it.
Nevertheless, the reviewer continues with pointing out that “References: Why some names are all in caps, others all in lower case?”
We checked the bibliography section and fixed names in lower case.
Also, this reviewer stated that the English level was “understandable but awkward, with many grammar errors”.
We did a complex language review resulting in fixing not only the examples pointed out by the reviewer, but also by editing dozens of minor grammar mistakes, which could have been found in the originally submitted text.
Reviewer 4 Report
This paper focused on survey-based approach to define the way of support of blockchain technology in avoidance false origin of IP prefixes or hijacking AS paths. A detailed survey has been conducted. All literature supported well to the content of the paper. For this, he presented two algorithms for Scheduling and Migration tasks. The study seems interesting and valuable. My comments are below for better enhancement:
- Figures need a visible and equal font size, and better if you can add EPS images for more clarity of your work.
- If you have some own contribution then please make a new section “Contribution of study” just after the related work section, else no need.
- Also, before the conclusion section, write the limitation of a defined survey of work.
- Minor typos and writing bugs can be solved.
Author Response
This reviewer states that “a detailed survey has been conducted. All literature supported well to the content of the paper. .. The study seems interesting and valuable.”
Similar to Reviewer 1, also here we are very pleased and thankful to this kind of positive feedback on our work.
Nevertheless, the reviewer continues by pointing out that “figures need a visible and equal font size, and better if you can add EPS images for more clarity of your work.”
All of these suggestions have been taken into consideration, and the resubmitted manuscript contains images in EPS format. Also, we changed the width of images to 10 cm.
The review continues with a suggestion, that we shall “before the conclusion section, write the limitation of a defined survey of work.”
We consider this to be very valuable advice, and the limitation of a defined survey work was added just before the conclusion section.
Last but not least, this reviewer also reminds that typos and minor grammatical errors should be fixed. As stated in the response to Reviewer 3, the grammar check has been done and multiple mistakes were fixed.